# Evaluation of a Real-Time Monitoring and Management System of Soybean Precision Seed Metering Devices

**Jicheng Zhang** [1], **Yinghui Hou** [1], **Wenyi Ji** [2], **Ping Zheng** [1], **Shichao Yan** [1], **Shouyin Hou** [2] **and Changqing Cai** [3,4,*]

1. College of Electrical and Information, Northeast Agricultural University, Harbin 150030, China
2. College of Engineering, Northeast Agricultural University, Harbin 150030, China
3. College of Electrical and Information Engineering, Changchun Institute of Technology, Changchun 130012, China
4. National and Local Joint Engineering Research Center for Smart Distribution Network Measurement, Control and Safe Operation Technology, Changchun 130012, China
* Correspondence: caicq@ccit.edu.cn; Tel.: +86-13-504-400-544

**Abstract:** Aiming at precise evaluation of the performance of soybean seed metering devices, a photoelectric sensor-based real-time monitoring system was designed. The proposed system mainly included a photoelectric sensor module for seeding signal collecting, Hall sensors speeding module, microcontroller unit (MCU), light and sound alarm module, human–machine interface (HMI), and other parts. The indexes of miss, multiples, flow rate, and application rate were estimated on the basis of seeder speed, seed metering disk rotation rate, photoelectric sensor signals, and clock signals. These real-time statistics of the seeding process were recorded by seeding management system. The laboratory results showed that the detection errors of seeding quantity of both big- and small-diameter soybeans were less than 2.0%. Miss and multiples index estimated by this system were 0.4% and 0.5% than that of seeding image monitoring platform (SIMP), respectively. In field tests, miss and multiples index can be used to evaluate the performance of seed metering device, and big-diameter seeds can be detected more precisely than small ones by these photoelectric sensors. This system can provide support for evaluation of working performance of seed metering devices and have a positive effect on seeding monitoring technology.

**Keywords:** seed metering device; monitoring system; photoelectric sensor; miss; multiples; flow rate; smart agriculture

## 1. Introduction

The precision seeding machine has been widely used in precision/smart agriculture recently. It plays an important role in crops growth, yield, and final production income [1]. At present, most precision seeders used are mechanical and pneumatic technology [2]. The phenomenon of miss seeding [3], multiples, and blockage have been reported to often occur [4,5]. However, detecting these problems promptly is difficult for users, which tends to lead to large-scale missed seeding and serious economic losses [3,6]. So, the real-time monitoring of seeding quantity and quality is necessary for modern precision seeding machines.

Especially in the Huang-Huai-Hai region of China, rotation of winter wheat and summer soybean cropping is the main mode of agricultural production. Due to the increase of straw retained after the wheat harvest without rotary tillage or plow tillage, no-tillage seeding mechanism possibly leads to the blockage of later soybean planters [7], which results in large-scale seedless and laborious replanting. Seeding monitor system is one of the important methods to solve with this issue. As we have known, there is no monitoring system for no-tillage seeding machines, in which there were more complex field conditions and difficulty to find out missed seeding. Meanwhile, statistical indexes for the seeding

process, such as seeding quantity, sowing rate, application rate, and planting density, have important monitoring value [8]. Therefore, the performance monitoring of seed metering devices under operational conditions needs to meet the requirements of crop seeding flow rate and its working efficiency [9].

Common monitoring techniques for seeding include photoelectric [10], machine vision [11], laser [1], and microwave detection [12] technology. The photoelectric monitoring method is widely used because of the characteristics of its high accuracy and low cost [3,13,14]. Karimi et al. evaluated that the infra-red (IR) detection technique was more suitable for non-contact sensing techniques for estimating of the seed flow rate than other investigated sensing methods [10]. Hao et al. set the speed of a seedling bed to 3, 4, and 5 km·h$^{-1}$ based on the seed metering performance monitoring system with a precision seeder with a photoelectric sensor. They obtained 94.4% monitoring accuracy with the system [15]. Zhang et al. also evaluated a soybean precision seeder monitoring device with a photosensitive sensor. They set the moving speed of the seeder in field operation to 3.6 km·h$^{-1}$ and 5.4 km·h$^{-1}$ and obtained a monitoring accuracy of more than 96% [16]. The methods and technology of getting more precise is one of the main development directions in seeding monitoring field.

Currently, data is an increasingly important factor in agricultural production, and agricultural machinery is a major vehicle for the acquisition and utilization of agricultural data [17]. There are a large number of studies about agricultural environmental data collection. However, only a limited number of statistical studies had been conducted on the work process of seeders, which led to single-machine data management deficiency and wide-area big data management issues. A real-time seeding monitoring system equipped on a precision seeder, which is a fundamental data collect unit, should not only be used for the seeding process, but also for the development level of agricultural informationization and digitalization [18]. Zhao et al. performed indirect measurement methods of seed and fertilizer operation amounts to calculate the relative errors [19]. Karimi et al. also designed a seeding monitoring system based on the infrared seed sensor, which could provide a calibration method and monitor the sowing process in the field [20]. The scatter plot of application-rate information was drawn by using an application rate measuring system equipped on a mechanical wheat seed–fertilizer drill machine, which could help to evaluate the dynamic process of fertilization and seeding to precisely measure application rate, and to achieve a variable rate application [21]. Along with the development of big data application, agricultural machinery working information is necessary to establish a cross-regional and national data system for extensive investigation and to provide production decisions [17]. So, on the basis of precise seed metering monitoring, it not only accurately detected abnormal seeding conditions of miss, multiples, and blocked seeding, but also recorded real-time seeding statistics and promptly reflected the seeding quality and working efficiency for users, thereby providing important production guidelines and economic benefits [22]. This kind of monitoring system with the characteristics of low-cost and efficiently will be one of the main scopes of precise agricultural machine research in the future.

In this study, a real-time monitoring and management system based on photoelectric sensors for soybean precision seed metering devices was presented, which could monitor hermetic seed tubes without a blind area and could trigger alarms for abnormal status. A statistical system was incorporated using moving speed, planting space, and sensor monitoring signals to calculate seeding quantity, seeding rate, and planting space in real time. The objective of this study was to evaluate the performance of seeding sensors and statistics systems, improving the level of seeding operation management.

## 2. Materials and Methods

### 2.1. System Components

Seed metering monitoring system was designed as shown in Figure 1. The proposed system mainly included a photoelectric sensors module for seeding signal collecting, Hall

speed monitor module, microcontroller unit (MCU), sound–light alarm module, human–machine interface (HMI), and communication parts. Hall sensors converted quantities of ground wheel rotation rate into pulse signal [23,24]. A human–machinery interface (HMI) was used to display data and for user input. An alarm circuit with sound and light and two types of prompt was designed to better transmit alerts to the users [25]. The required power supply was provided by a tractor battery after DC 12 V–DC 5 V voltage reduction.

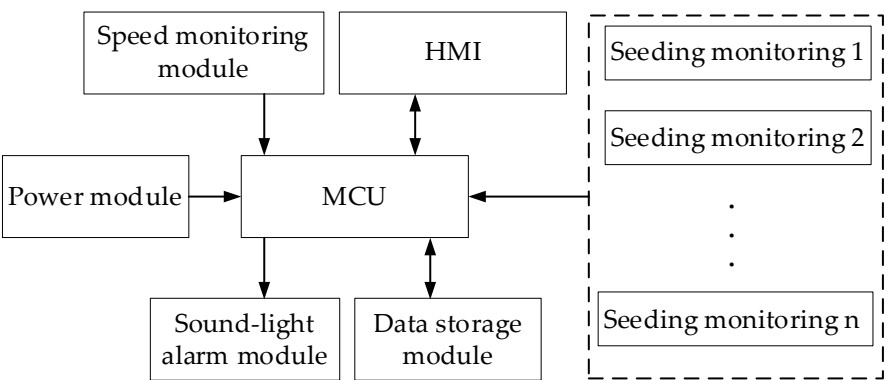

**Figure 1.** Structure diagram of the monitoring system of the seeder.

*2.2. Working Principle*

A convert circuit made the intermittent seed flow signals into electrical pulse signals. The time interval $\Delta t$ (s) between two adjacent pulse signals reflected the time difference of falling seeds. Through time interval $\Delta t$ and the moving speed of seeder tractor $v_{tractor}$ (m·s$^{-1}$), the actual grain planting space can be estimated.

In accordance with Chinese National Standard as GBT 6973–2005 (Chinese National Standard, 2005), the theoretical and actual grain spacing grain had to be compared. The formula judging the occurrence of multiples or miss seeding was as follows:

$$\begin{cases} v_{tractor}\Delta t \leq 0.5\bar{d}\,(multiplied\ seeding) \\ v_{tractor}\Delta t > 1.5\bar{d}\,(missed\ seeding) \end{cases} \tag{1}$$

where $\bar{d}$ was the theoretical planting space, cm.

To increase the accuracy of rotation measurement, some magnetic steels were installed on the ground wheel of seeder. Assume the number of magnetic steels was $Z$. Hall sensors were used to collect the rotation pulse signals of the ground wheel and to record the pulse number $N$ in the speed measuring period $T$ (s).

The slip ratio of the ground wheel was key to ground wheel speed measurement. The equation was:

$$\sigma = \frac{S - \pi D n}{\pi D n} \times 100 \tag{2}$$

where $\sigma$ was the slip ratio of the ground wheel, %. In general, its range was 0.05–0.12; $S$ was the actual forward distance of the ground wheel, m; $D$ was the diameter of the ground wheel, m; $n$ was the number of rotations of the ground wheel during $S$ m.

The seeder speed $v_{tractor}$ (m·s$^{-1}$) was calculated as follows:

$$v_{tractor} = \frac{N}{Z}\frac{\pi D(1 + \sigma)}{T} \tag{3}$$

The rotation rate of the seed metering disk $\omega$ (r·min$^{-1}$) was calculated by the equation:

$$\omega = 60i\frac{N}{Z}\frac{(1 + \sigma)}{T} \tag{4}$$

where *i* was the transmission coefficient between the wheel and the seed metering rotation rate.

Assume that the seed metering device had *m* seed metering holes, the seed metering quantity in per min was *Q*, which was expressed as follows:

$$Q = m\omega = 60i\frac{mN}{Z}\frac{(1+\sigma)}{T} \tag{5}$$

Then, from Equations (3)–(5), the actual planting space $\Delta t'$ (s) of two adjacent seeds was approximately calculated as follows:

$$\Delta t' = \frac{60}{Q} = \frac{ZT}{i \times m \times N(1+\sigma)} \tag{6}$$

Subsequently, $\Delta t'$ derived from Equation (6) and the time interval of two falling seeds $\Delta t$ monitored by the seed metering sensor can be used to cross-validate the monitoring system for multiples and miss seeding, thereby improving the sensitivity of the system in monitoring seeding status.

The seed metering number $Q_i$ (1, 2, … , n) of each seed metering device in period *t*(s) can be accumulatively obtained through the signals of the seed metering sensors, and the total quantity of seed metering $Q_{total}$ (grain) and the sowing rate of the seeder $v_{sow\_rate}$ (grain·s$^{-1}$) were listed as:

$$Q_{total} = \sum_{1}^{n} Q_i \tag{7}$$

$$v_{sow\_rate} = Q_{total}/t \tag{8}$$

The working area of the seeder was the product of the working distance and working width of the seeder. The application rate of the seeder $Q_{average}$ (grain·m$^{-2}$) can be deduced by the below equation:

$$Q_{average} = \frac{Q_{total}}{v_{tractor} \times t \times L} \tag{9}$$

where *L* was the working width of the seed metering devices, m. $Q_{average}$ reflected the working efficiency of the seed metering device and can estimate planting density [26,27].

### 2.3. Software Working Flow Design

The flow diagram of the monitoring system included system initialization, timing interruption, signal acquisition, sound–light alarm, and seeding statistics. The system software flow was shown in Figure 2.

The program was first initialized after the normal activation of the monitoring system, and its self-checking function was executed. The system automatically checked for the occurrence of any abnormality.

After self-checking, three types of data had to be input by HMI. The first was agronomic requirements, for example, pre-set seeding quantity and planting space. The second type was physical property of the seed tractor, such as ground wheel diameter, the parameters of slip ratio, transmission coefficient for ground wheel speed, and seed metering amount, respectively. The last one was alarm threshold. The range of threshold could be set to adjust the sensitivity of this system. If the range was smaller, more miss and multiples were detected and the more precise the monitoring system was. However, it would frequently lead to alarms to affect normal work. If the range was bigger, much abnormal status might be lost at seeding process. So, a suitable threshold range had to be input according to actual requirements. Before field working, the alarm threshold had to be tested at first. Then, the alarm worked when time interval $\Delta t$ between two adjacent pulse signals of seeders were out of the range of threshold based on Formula (1). Otherwise, the buzzer did not alarm.

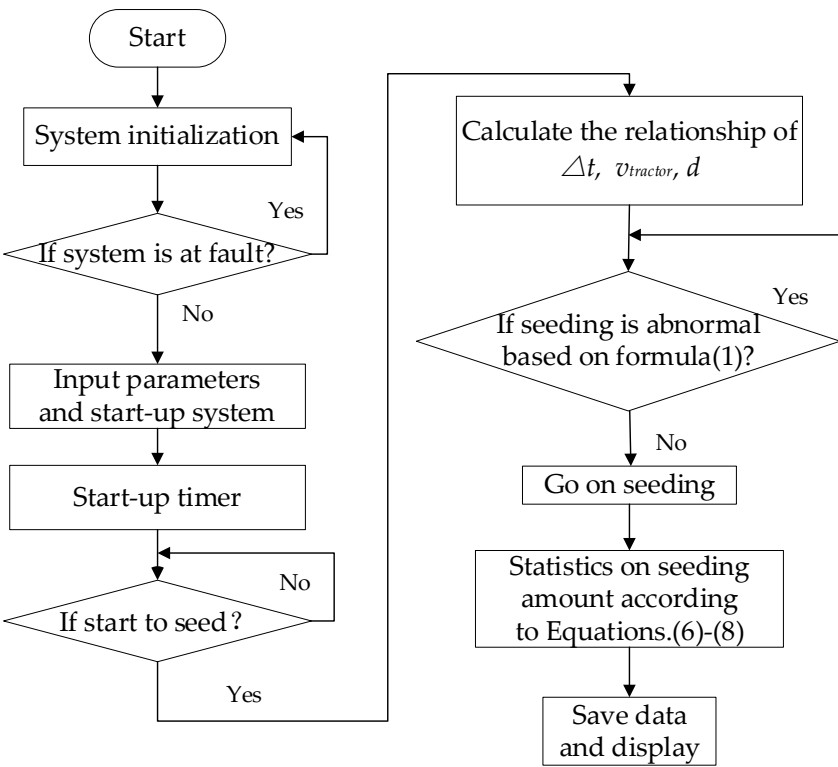

**Figure 2.** Monitoring system flow diagram.

After the start of the monitoring process and after receiving data from sensors, the microcontroller performs relevant calculations. It simultaneously monitored seed flowing, recorded the time interval between adjacent seeds, and detected miss seeding and multiples rate on the basis of the Equation (6). The system can also analyze whether the alarm coefficient exceeded thresholds according to Formula (1). Through the time interval $\Delta t$ and the tractor moving speed $v_{tractor}$, the actual planting space was calculated. If actual planting space was less than half theoretical planting space, the system judged it was multiples. If actual planting space $d$ was more than one and a half times of theoretical planting space, the status of miss seeding may occur and the sound–light alarm worked.

However, when the seeder should stop planting at the edge of the planting rows, the ground wheel speed was much smaller than usual speed and no seeds passed seed sensors, and the alarm was not needed.

Taking into account the input signals received from each seed tube and pre-registered settings by users, the seed metering statistics, such as total seeding quantity, seeding rate, miss index, and multiples index, were stored on the memory card and were also transferred through the serial port to HMI. They can provide performance information for users. Finally, the worker shut up the monitoring system and kept the data in the memory card for later analysis and decision.

### 2.4. Seed Metering Monitoring Sensor

Low-power infrared-emitting diodes and phototransistors are widely used for their characteristics of rapid response, high transmission efficiency, and high monitoring sensitivity. The light beam was concentrated, and its dust-penetrating ability is stronger than that of visible light.

Photoelectric sensors module was used by covering the seed metering tube with the emitting and receiving ends of three pairs of infrared light-emitting diodes and phototransistors. An infrared-emitting diode with a diameter of 10 mm emit infrared light with a wavelength of 850 nm and an emission angle of 60° [16]. The monitoring sensor was designed as a cuboid structure with a length, width, and height of 45 mm × 23 mm × 40 mm,

respectively, to enable the emitted light to cover the entire monitoring device without a blind area (Figure 3). In this study, after the finalization of the circuit and the design of a shell for the seed sensor, the shells were produced using a 3D printer. The material used should not let ambient light disturb the sensor inside, and these sensors should be installed where soybeans flew vertically to improve measurement accuracy.

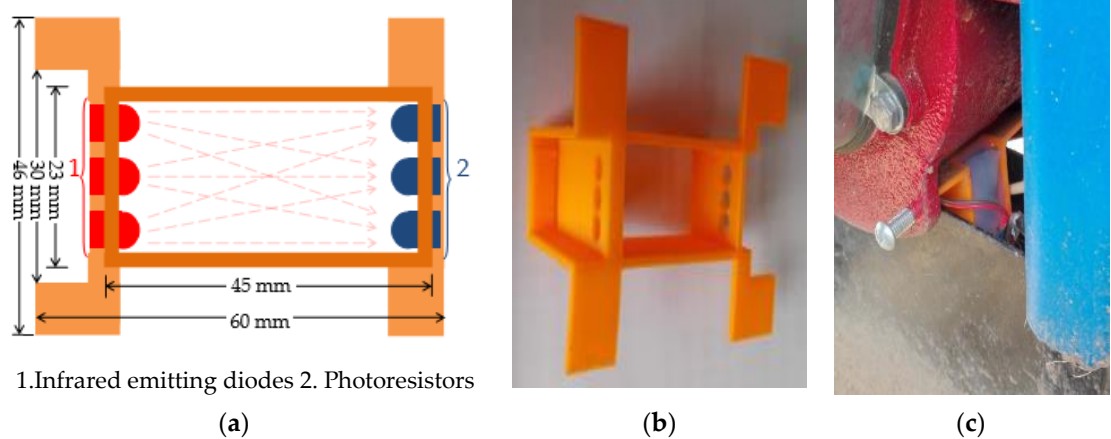

1.Infrared emitting diodes 2. Photoresistors

(**a**)  (**b**)  (**c**)

**Figure 3.** Photoelectric sensor design. (**a**) Schematic diagram; (**b**) a sensor shell produced by a 3D printer; (**c**) sensor installation.

During the seeding operation, seeds entered the seed metering tube and continuously passed through the monitoring sensors installed at the bottom of the seed metering devices (Figure 4).

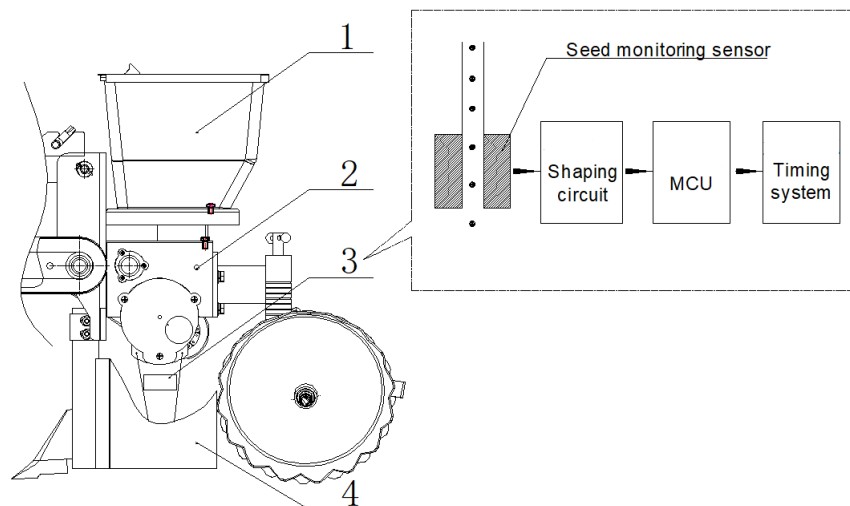

**Figure 4.** Schematic diagram of monitoring sensor position. (1). Seed box; (2). spoon-type seed metering device; (3). monitoring sensor; (4). ditching device side panel.

As shown in Figure 5, the seed metering monitoring circuit was primarily used to obtain information when seeds passed through the monitoring device. The monitoring circuit was primarily composed of an infrared-emitting diode, a phototransistor, a voltage comparator, and other devices. LM324 was used as a voltage comparator to consider the reliability and stability of the captured data. A sliding rheostat R4 was used to adjust the sensitivity of the sensor on the basis of the different precision required by the variety of seeds.

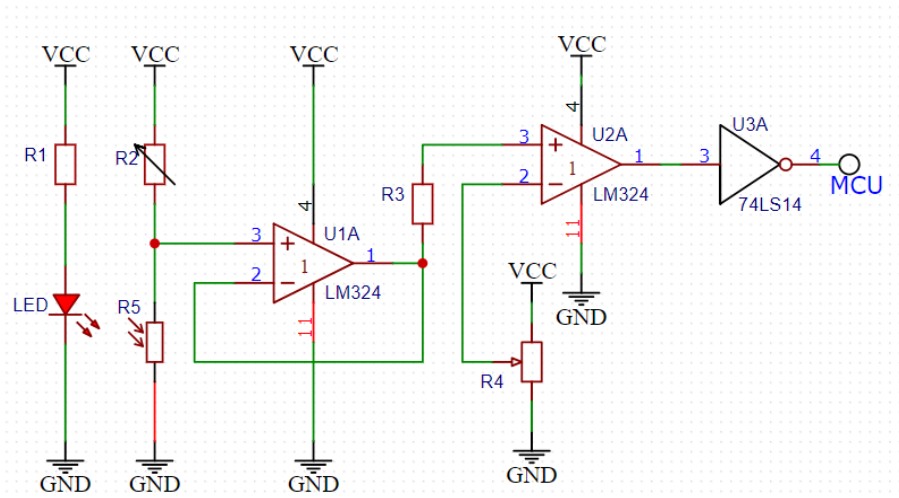

**Figure 5.** Sensor circuitry principal diagram.

When no seed passed through, all photodiodes were exposed to light at the same time. When there was a seed passing through, the photosensitive diode could not receive the light. At this time, the corresponding transistor was in the cut-off state, making the collector output a high level. Schmidt trigger 74LS14 was used to filter, reduce noise, and shape the signal at the collector output end to obtain square wave pulse, which was convenient for the calculation and processing of MCU.

## 3. Results

### 3.1. Monitoring Test for Sensitivity

To evaluate the monitoring sensitivity of the seed metering monitoring sensor, laboratory tests were conducted with two soybean varieties with different diameters [28]. The variety of small diameter was Dongnong 46, and its 100-grain weight was approximately 21.78 ± 0.40 g, with a diameter of 4.5 ± 0.2 mm. The other variety of large diameter was Dongnong 252, and its 100-grain weight was 28.28 ± 0.52 g, with a diameter of 5.5 ± 0.3 mm. Some researchers also focused on the sowing of various seeds with these types of physical properties [13].

The number of seeds was counted when seeding speed was stable. Seed metering devices 1 to 5 were filled with small seeds, and seed metering device 6 was filled with large seeds (Figure 6). The monitor results were recorded in Table 1.

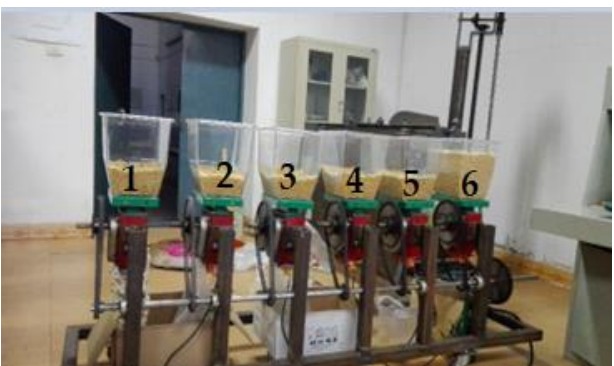

**Figure 6.** Seeding test platform.

**Table 1.** Monitoring data for soybeans with different diameters.

| Items | Small Seeds | | | | | Big Seeds |
|---|---|---|---|---|---|---|
| | Sensor 1 | Sensor 2 | Sensor 3 | Sensor 4 | Sensor 5 | Sensor 6 |
| Seeds quantity monitored/grain | 11,473 | 11,553 | 11,472 | 11,432 | 11,519 | 11,000 |
| Seeds weight/g | 2540.00 | 2560.12 | 2539.24 | 2536.13 | 2561.63 | 3062.85 |
| Seeds quantity calculated/grain | 11,660 | 11,753 | 11,657 | 11,642 | 11,760 | 10,829 |
| Error/% | 1.6 | 1.7 | 1.6 | 1.8 | 2.0 | 1.6 |

As indicated in Table 1, the error of the monitoring sensors for the soybean variety with a small diameter was higher than that for large-diameter soybeans. This result was mostly attributed to the large-diameter soybean screening out infrared beam for a slightly longer time in the sensors. The signal was transmitted to the single-chip calculator, where it could be effectively identified and counted. It can be seen that the monitoring errors of both of them were less than 2.0% for both big and small soybeans. These errors may be caused by multiples in which two adjacent and nearly overlapping soybeans fell down at the same time. This phenomenon could not be precisely detected by this kind of sensor.

### 3.2. Static Monitoring Test

To evaluate the monitor performance, seed metering devices were installed on a seeding image monitor platform (SIMP) (Figure 7) to compare with the photoelectric sensor monitoring system [29,30].

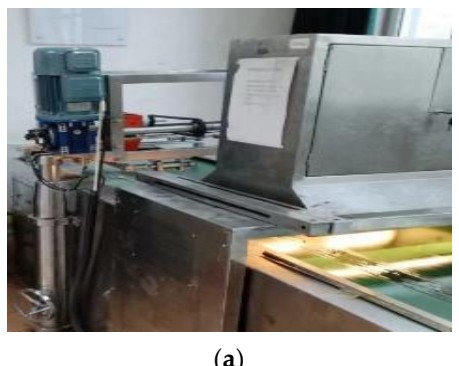
(**a**)

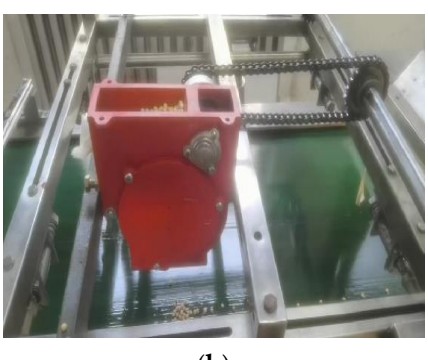
(**b**)

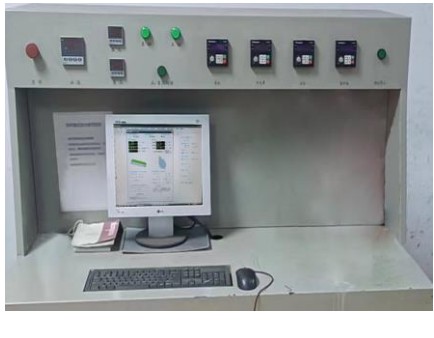
(**c**)

**Figure 7.** Seeding image monitor platform (SIMP). (**a**) Camera-monitoring platform; (**b**) seed metering speed control bracket; (**c**) image-monitoring interface.

Soybean seeds (variety: Dongnong 46) were used. A "Shuang Fu" brand mechanical precision seed metering device, which had a vertical spoon wheel type with 33 holes, was selected. The inverter was used to control three rotation speeds of the seed metering wheel (9, 18, 27 (r·min$^{-1}$)). The inverter speed was set to repeat five times at each rotation speed. The test data of multiples, miss of SIMP, and this monitoring system were recorded when speed was stable, and the average value of the five tests was used as the test result in Table 2. Flow rate was estimated by Equation (8), and application rate was estimated by Equation (9).

As shown in Table 2, the errors of the miss index between monitoring system and SIMP were less than 0.4% at three rotation speeds, and the errors of multiples index between them were less than 0.5%. This monitoring accuracy was nearly equal to article [1] based on laser sensors, in which the average monitoring error of the seeding qualified rate was less than 0.5% on a simulated seeder at a forward speed of 5–10 km·h$^{-1}$.

**Table 2.** Statistics results of static experiment.

| Speed/m·s⁻¹ | Disk Rotation Rate/r·min⁻¹ | Miss Index/% | | | Multiples Index/% | | | Flow Rate/Grain·s⁻¹ | Application Rate/Grain·m⁻² |
|---|---|---|---|---|---|---|---|---|---|
| | | SIMP | Monitor System | Error | SIMP | Monitor System | Error | | |
| 0.50 | 9.00 | 1.20 | 1.10 | 0.10 | 2.30 | 2.10 | 0.20 | 28.31 ± 2.40 | 16.18 |
| 1.00 | 18.00 | 1.80 | 1.60 | 0.20 | 3.10 | 2.80 | 0.30 | 56.26 ± 5.52 | 16.07 |
| 1.50 | 27.00 | 2.50 | 2.10 | 0.40 | 4.40 | 3.90 | 0.50 | 85.17 ± 6.82 | 16.22 |

The seeding rate and planting space had a great influence on the monitoring accuracy of the sensor. After analysis, the results were found that the errors increased with an increase of rotation speed of seed metering disk. When flow rate reached a high value of 85.17 grain·s⁻¹, the error of miss and multiples index of this monitoring system were the biggest of these three speeds. Application rates were nearly unaffected, calculated by Equation (9).

*3.3. Field Dynamic Monitoring Test*

To test the monitoring system in the field [8,31], no-tillage machine seeding monitor tests were conducted on wheat stubble in Yongcheng City, Henan Province. Small-diameter soybean variety (Zhengdou 04024) was used in 2016, and big-diameter soybean (Hedou 33) was used in 2019. The 100-grain weight of Zhengdou 04024 was 21.3 ± 0.09 g, with the diameter 4.9 ± 0.4 mm after sampling measurement. The 100-grain weight of Hedou 33 was 24.2 ± 0.11 g, with the diameter 5.7 ± 0.5 mm.

Experimental field was without rotary tillage or plow tillage. A John Deere 954 tractor equipped with a six-row seeder had a working width of 3.80 m (Figure 8). "Shuang Fu" brand mechanical precision seed metering devices, which had vertical spoon wheel type with 33 holes, were used. Sowing rate was adjusted by mechanical transmission device of gears. Gear I, II, and III corresponded to the planting space of 5 cm, 8 cm, and 10 cm. The tractor ran at a speed about 5 km·h⁻¹ (1.38 m·s⁻¹) in these two years. The higher speed of the tractor led to more axis dropped out. A straw-mulching clearing device was equipped on a no-tillage planter in 2019, which was helpful to decrease blockage of seeding tubes. However, the complex field condition especially, in a no-tillage field, most affected the performance of these optical sensors. It is impossible to completely eliminate missing inspection. This was one of main obstacles for improving the accuracy of the monitoring system [20].

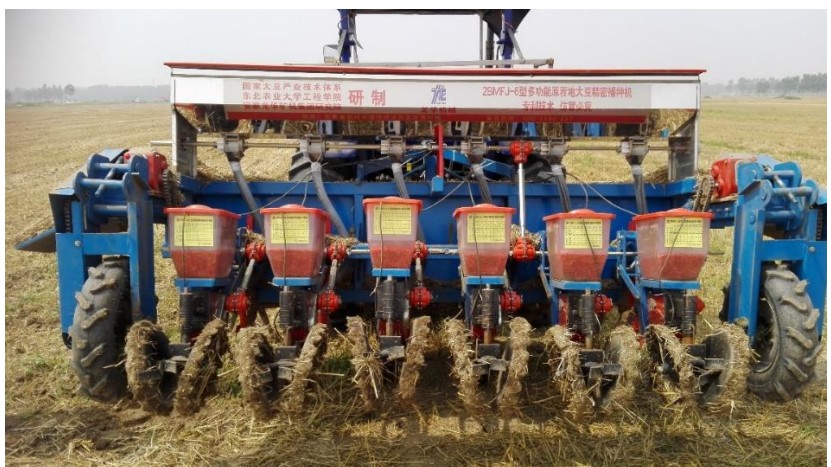

**Figure 8.** Field test.

The monitoring sensors were installed under the seed metering devices (Figure 3). Sowing rate (estimated by Equation (8)) and application rate (estimated by Equation (9)) were recorded in Table 3. Abnormal phenomenon records were shown in Table 4.

**Table 3.** Dynamic experimental results of the digital monitoring system.

| Gear No. | Planting Space/cm | Small Diameter (2016) | | Big Diameter (2019) | |
|---|---|---|---|---|---|
| | | Sowing Rate/Grain·s$^{-1}$ | Application Rate/Grain·m$^{-2}$ | Sowing Rate/Grain·s$^{-1}$ | Application Rate/Grain·m$^{-2}$ |
| I | 5.0 | 160.43 | 32.29 | 172.2 | 32.84 |
| II | 8.0 | 100.58 | 20.24 | 109.18 | 20.82 |
| III | 10.0 | 80.73 | 16.25 | 85.48 | 16.30 |

**Table 4.** Abnormal phenomenon statistics in field tests.

| Gear No. | Small Diameter (2016) | | | | | | Big Diameter (2019) | | | | | |
|---|---|---|---|---|---|---|---|---|---|---|---|---|
| | Miss | Multi Ples | Blockage | Quantity/ Grain | Miss Index/% | Multiples Index/% | Miss | Multi Ples | Blockage | Quantity/ Grain | Miss Index/% | Multiples Index/% |
| I | 1986 | 2134 | 5 | 63,650 | 3.12 | 3.36 | 1590 | 1794 | 4 | 52,360 | 3.04 | 3.42 |
| II | 1846 | 1952 | 5 | 63,210 | 2.92 | 3.09 | 1443 | 1726 | 2 | 53,009 | 2.72 | 3.26 |
| III | 1583 | 1722 | 6 | 63,300 | 2.50 | 2.72 | 1237 | 1546 | 3 | 54,132 | 2.29 | 2.86 |

As shown in Table 3, planting space continuously increased with an increase of seed metering gear. Meanwhile, sowing rate and application rate (planting density) decreased. Through analysis, flow rate and application rate in 2016 were a little smaller than 2019 (Table 3). Meanwhile, the percent of multiples in 2019 was a little larger than 2016 (Table 4). That meant a greater percentage of big-size soybeans was detected under the same theoretical sowing rate. Through the monitor, miss index of big-size soybean was a little lower than small-size ones. Miss index and multiples index continuously decreased with a decrease of metering devices speed, and it did not exceed 3.12% and 3.42%, respectively. It can be seen that this kind of soybean seed metering device had a good seeding performance at these working conditions.

After cleaning out the impurities, the number of blockage alarms in 2019 was obviously lower than 2016 (Table 4).

After the field test, three major phenomena had to been stated: (1) Seeds in the seed boxes contained many impurities, which affected the filling effect of the seed metering holes. Light and sound alarms occurred when miss seeding. (2) The uneven ground in the test field prevented the tractor from moving at a uniform speed. When seed metering speed was occasionally fast and slow, the statistical value of seed metering fluctuated a lot. (3) The dust amount of no-tillage seeding in the wheat stubble field was considerable, which affected the accuracy of the monitoring system. The self-clean or dust-reduction device should be employed on seeders.

## 4. Conclusions

(1) The seed metering monitoring sensor used three pairs of infrared light-emitting diodes and phototransistors as the emitting and receiving ends of the photoelectric sensor, which could achieve the blind area free monitoring of the soybean seed metering tube. This sensor was simple, practical, and low-cost.

(2) In laboratory tests, the monitoring of soybeans with big and small diameters was basically consistent, and the monitoring errors of seeder quantity was less than 2.0%. The errors of miss and multiples index between monitoring system and SIMP were less than 0.4% and 0.5%, respectively, at three rotation speeds. This demonstrates that this kind of sensors have a good monitoring performance.

(3) This monitoring system can evaluate the precision of the performance of seed metering device in no-tillage machine seeding tests and verify that the big-diameter seeds can be detected more precisely than small ones.

This real-time monitoring and management system of soybean precision seed metering devices was helpful to prevent large-scale miss seeding and collect and record the quantity and quality information of seeding process.

**Author Contributions:** Conceptualization, W.J. and S.H.; methodology, J.Z.; software, S.Y.; validation, P.Z. and Y.H.; formal analysis, J.Z.; data curation, J.Z.; writing—original draft preparation, P.Z. and C.C.; writing—review and editing, J.Z. and C.C.; project administration, J.Z.; funding acquisition, J.Z. All authors have read and agreed to the published version of the manuscript.

**Funding:** This research was funded by the National Science Foundation of Heilongjiang Province, grant number LH2020E002, the Department of Science and Technology of Jilin Province, grant number 20200402116NC and Soybean Industrial Technology System of China, grant number CARS-04-PS27.

**Data Availability Statement:** Not applicable.

**Conflicts of Interest:** The authors declare no conflict of interest.

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
