# Peer review of "Evaluation of a Real-Time Monitoring and Management System of Soybean Precision Seed Metering Devices"

_agronomy, doi:10.3390/agronomy13020541_

Round 1

Reviewer 1 Report

The manuscript’s (MS) topic deals with the development of photoelectric seeding monitoring sensors. A thing to take into account is that similar systems have already been presented in previous publications. 

The development of a monitoring and management systems for a seeder can be interesting and necessary to achieve the objective, but is not a new scientific evidence enough to be published in a high standard international scientific journal.

I list the following comments and/or concerns to provide some feedback to the authors: 

1) The literature review of the manuscript is insufficient. There are many articles related real time monitoring systems for seeders. The authors should have to evaluate more literatures on this subject. The second important issue is the usage of incorrect technical terms in the manuscript. For example the authors used the "misseeding" and "reseeding" terms for multi or missed seedlings. The correct terms for these events are "multiple index" and "miss index", respectively. I advise the authors to detailing evaluations on the following article and ISO standards on precision seeders. This is an obligatory for the use of appropriate technical terms and for the evaluation of precision seed drills in accordance with the standards.

Kachman S D; Smith J A (1995). Alternative measures of accuracy in plant spacing for planters using single seed metering. Transactions of the ASAE, 38(2), 379–387

2) Line 36-37: When I read the first sentence of Introduction, I thought that a monitoring system for high speed seeders was developed in this manuscript. However, the test was conducted for normal forward speed (line 232). This sentence will be reason for misunderstanding and should be corrected.

3) As I mentioned above there are so many article on monitoring sensors for seeders, therefore a more comprehensive literature review should be done and the superiority of this system over existing (published) systems should be expalined? 

Equation 8: No need for this equation. 

Line 174: is the 7 years long period for publishing a research results isn't it?!!

Lines 177-180: The standard deviations should be presented together with the mean values.

Table 1: There is no result for different variety of the seed, only for different sensors. What is the difference between the sensors and where is the results for different variety of seed?

Line 209-210: If the values are average, the standard deviation should be added to average values.

Table 2: There is no results for errors mentioned in lines 210-213.

Line 224-225: what is “no-tillage machine seeding monitoring”? I think a detailed revision for technical terms are needed.

Line 228: The seed variety for field test is different from static monitoring test and authors claim that the field test was conducted for verification. This is not acceptable!!

Line 239-240: “In the field experiment, the rates of miss-seeding and reseeding of the seeders increased” How did you compared? No results for miss-seeding and reseeding for static tests and reseeding for dynamic tests, only for miss-seeding was presented in Table 3.

Author Response

Dear reviewer,

Thanks for your comments. It really give us a lot of help. We have updated our manuscript as your suggest.

Reviewer 2 Report

Dear editor,

The manuscript "Evaluation of a real-time monitoring and management system of soybean precision seeder" was submitted to Agronomy. It's a nice research, but I have some concerns and my decision is major revision, finally I recommend to authors that improve the quality of this nice paper as following.

1.      The results should be stated more accurately in the abstract. This will be of great help to the readers in the future and has an effective role in attracting the readers.

2.      Please talk more about the results and conclusions in the abstract to improve the quality.

3.      There are repeated words between the title and keywords, please revise them, please be careful.

4.      In the introduction, try to write more related to the necessity and background of the research.  It’s very important.

5.        The materials and methods are well written. congratulations

6.      I can’t find discussion, please be careful. 

7.      One of the significant concerns is that the authors should carefully develop a discussion section to talk about the significance, shortages or advantages of the methods you proposed, the reliability and meaning of your results (compared to other related studies) etc.

8.      I can’t find conclusion, please be careful. 

9.      In the references, it is recommended to use new and updated research. At least 40% of them should be related to the last four years.

10.      Finally, I checked plagiarism detection of this research and the similarity is 15%, please checked attached file.

Author Response

(The authors gave the same response as above.)

Round 2

Reviewer 1 Report

The authors' responses to most of my comments are satisfactory. They have improved the manuscript by incorporating most of my suggestions and corrections.

There is only one issue that; I still believe that the seed variety used for field test should be the same with the laboratory test for verification of monitoring system. If you carried out field test only for see the monitoring system performance in field condition, it is OK but it  it wroten in line 275 that carried out for verification. This is not acceptable!! The verification test have to be repeated for the same operation conditions (such as speed and seed spacing) and seeds.

Author Response

The authors' responses to most of my comments are satisfactory. They have improved the manuscript by incorporating most of my suggestions and corrections.

There is only one issue that; I still believe that the seed variety used for field test should be the same with the laboratory test for verification of monitoring system. If you carried out field test only for see the monitoring system performance in field condition, it is OK but it  it wroten in line 275 that carried out for verification. This is not acceptable!! The verification test have to be repeated for the same operation conditions (such as speed and seed spacing) and seeds.

Response: Due to regional differences, we did not choose different soybean varieties in laboratory and field tests. We have modified the manuscript for testing and not for verification according to your suggestion. In the later tests, we will keep this in our mind. Thank you very much for your correction of the paper.

Reviewer 2 Report

Dear editor, 

the quality of manuscript is suitable or publication. 

Regards

Author Response

Thanks for your correct of our manuscript. Your guide let us change and improve the manuscript a lot. We really appreciate your help.